# The Role of the IL-23/IL-17 Pathway in the Pathogenesis of Spondyloarthritis

**DOI:** 10.3390/ijms21176401

**Published:** 2020-09-03

**Authors:** Hiroyuki Tsukazaki, Takashi Kaito

**Affiliations:** Department of Orthopedic Surgery, Osaka University Graduate School of Medicine, 2-2 Yamadaoka, Suita, Osaka 565-0871, Japan; tsukazaki.hiroyuki@gmail.com

**Keywords:** Spondyloarthritis, ankylosing arthritis, psoriatic arthritis, interleukin-17, interleukin-23

## Abstract

Spondyloarthritis (SpA) is a subset of seronegative rheumatic-related autoimmune diseases that consist of ankylosing spondylitis (AS), psoriatic spondylitis (PsA), reactive spondylitis (re-SpA), inflammatory bowel disease (IBD)-associated spondylitis, and unclassifiable spondylitis. These subsets share clinical phenotypes such as joint inflammation and extra-articular manifestations (uveitis, IBD, and psoriasis [Ps]). Inflammation at the enthesis, where ligaments and tendons attach to bones, characterizes and distinguishes SpA from other types of arthritis. Over the past several years, genetic, experimental, and clinical studies have accumulated evidence showing that the IL-23/IL-17 axis plays a critical role in the pathogenesis of SpA. These discoveries include genetic association and the identification of IL-23- and IL-17-producing cells in the tissue of mouse models and human patients. In this review, we summarize the current knowledge of the pathomechanism by focusing on the IL-23/IL-17 pathway and examine the recent clinical studies of biological agents targeting IL-23 and IL-17 in the treatment of SpA.

## 1. Introduction

In 1974, the concept of a group that was termed seronegative spondarthritides was first introduced [1] and is now known as Spondyloarthritis (SpA). These heterogeneous rheumatoid-related diseases consist of ankylosing spondylitis (AS), psoriatic spondylitis (PsA), reactive spondylitis (re-SpA), inflammatory bowel disease (IBD)-associated spondylitis, and unclassifiable spondylitis. These diseases share various clinical manifestations including sacroiliac arthritis, spinal arthritis, peripheral arthritis, enthesitis, and extra-articular forms (uveitis, inflammatory bowel disease, and psoriasis [Ps]), and the clinical commonality led to the grouping of these diseases. In addition, after the discovery of IL-23 and IL-17-producing Th17, it was revealed that molecular pathomechanisms were also shared among the different types of SpA.

The pro-inflammatory cytokines IL-23 and IL-17 play an important role in activating the immune response in the host defense against pathogens and maintaining barrier functions of mucosal surfaces. Over the past several years, genetic, experimental, and clinical evidence that SpA was triggered by pathological activation of the IL-23/IL-17 axis has accumulated. Inflammation at the sites of tendon insertion into bone is one of the characteristics of SpA, and it distinguishes SpA from other rheumatoid diseases. Recent studies have suggested the involvement of IL-23 and IL-17 in producing cells with enthesitis. In this review, we focus on the current knowledge of the role of the IL-23/IL-17 pathway in the pathogenesis of SpA and summarize the results of recent clinical trials targeting IL-23 and IL-17 in the treatment of SpA.

## 2. The Concept of SpA

In the 1970s, several diagnostic criteria were proposed to define patients with a specific subtype of SpA, such as the modified New York criteria for AS [2,3]. However, these criteria had inherent limitations since they focused on only spinal symptoms. In 1990, Amor et al. proposed the first set of comprehensive classification criteria for the entire group of SpA conditions, which enables patients to be diagnosed with SpA through peripheral manifestations [4]. Comprehensive criteria similar to those established by Amor et al. were also proposed by the European Spondyloarthropathy Study Group (ESSG) in 1991 [5].

Currently, SpA patients are divided into two subtypes based on their predominant clinical presentation: axial SpA and peripheral SpA. Spinal symptoms are predominant in the former subtype, whereas peripheral arthritis is predominant in the latter subtype, with some overlap between these two groups. In addition, the term axial-SpA includes both patients who have already developed structural damage (radiographic axial SpA, also termed ankylosing spondylitis [AS]) and patients who have experienced only inflammation without bone changes, termed non-radiographic axial SpA, which may be detectable by magnetic resonance imaging (MRI) (Figure 1). To meet the need to establish new criteria for classifying non-radiographic axial SpA, the Assessment of SpondyloArthritis International Society (ASAS) conducted a large cross-sectional study, resulting in the ASAS criteria for axial SpA and peripheral SpA [6,7]. Important advances in the ASAS criteria included the use of MRI capable of detecting sacroiliac inflammation before radiographic changes could be confirmed with plain radiographs.

## 3. Pathogenesis

### 3.1. Genetic Background

Human MHC class Ι, also referred to as HLA, belongs to the cell surface proteins that are present on all nucleated cells and platelets. MHC class Ι presents small antigen peptides to the T cell receptor (TCR) of cytotoxic T lymphocytes (CTLs), playing a pivotal role in the immune system [8].

HLA-B27, one of the proteins belonging to MHC class Ι, was first reported to associate with AS in 1973 and is now considered the essential genetic factor in the pathogenesis of SpA [9]. The varying prevalence of HLA-B27-positive populations among ethnicities seems to contribute to the different epidemiology of SpA [10]. The high HLA-B27 positivity (90–95%) in AS patients suggests that HLA-B27 is strongly related to axial SpA. In contrast, the positivity is lower (22–36%) in PsA in which peripheral joints are mainly affected [11,12]. In addition, HLA-B27-positive PsA patients had a higher risk of axial involvement than did HLA-B27 negative PsA patients [13]. However, the fact that only 2–10% of an HLA-B27-positive population develops AS [11] suggests that the disease-developing mechanisms other than HLA-B27 contribute to the pathogenesis of axial SpA.

In the past decade, genome-wide association studies (GWAS) identified *ERAP1* (coding for endoplasmic reticulum aminopeptidase1 [ERAP1]) as a risk factor for AS and PsA [14,15]. ERAP1 is one of the aminopeptidases expressed on the endoplasmic reticulum. ERAP1 takes part in the process of trimming peptides in the endoplasmic reticulum (ER) to 8–10 amino acids to present an antigen by MHC class Ι molecules, such as HLA-B27 [16]. It is presumed that ER stress caused by HLA-B27 and ERAP1 may trigger the activation of the IL-23/IL-17 pathway [17]. HLA-B27 has a predisposition for misfolding, and the accumulation of the improperly folded HLA-B27 in the ER interferes with ER function, which can lead to ER stress. ERAP1 polymorphisms can also affect the function of antigen presentation by the ER, resulting in ER stress.

Additionally, the SNP of the IL-23 receptor (IL-23R) has been reported to be a risk factor for AS and PsA in GWAS studies [14,18]. In addition, variants of STAT3 and Tyk2, which are downstream targets of IL-23, have also been reported to be associated with AS and PsA [14,19,20].

### 3.2. Mechanical Stress

Both animal and clinical studies suggest the involvement of mechanical strain in the development of inflammation and bone formation at the enthesis.

Higher involvement of the lower limbs than the upper limbs for enthesitis [21,22] suggests that a higher load in the lower limbs may be related to the development of enthesitis [21,23]. Primarily in the spine of patients with AS, entheseal inflammation and subsequent new bone formation often occur at the anterior longitudinal ligament, which bears a higher load. In support of this concept, tail suspension in the collagen antibody-induced arthritis (CAIA) model mice attenuated Achilles tendon enthesitis and led to a decrease in osteophyte formation [24]. These results suggest the involvement of mechanical stress on the development and/or progression of AS though the exact molecular mechanisms have not been clarified.

### 3.3. Dysbiosis

The microbiome that is resident in the intestine of mammals plays an important role not only in the regulation of nutrition but also in the adjustment of immune systems. Gut microbiota influences the balance between Th1, Th2, and Th17, which are essential in the host defense [25]. Once some environmental- or host-related factors alter the configuration of the Th cells, abnormal microbiome structure, referred to as dysbiosis, can induce several autoimmune diseases [26,27]. The HLA-B27 transgenic rat, which is a model of SpA, remains healthy in a germ-free environment due to the absence of IL-17-producing Th17 cells [28]. However, they developed SpA when exposed to commensal bacteria, such as segmental filamentous bacteria (SFB) [28,29]. The results suggest that the interactions between HLA-B27 and the microbiome are relevant to the pathogenesis of SpA. Clinical evidence has also been reported. Colonoscopies identified microscopic gut inflammation in 46.2% of the patients with SpA. Histological gut inflammation was correlated with the disease activity [30] and bone marrow edema in sacroiliac joints [31], supporting the involvement of dysbiosis in SpA. Interestingly, the effects on bacterial diversity differ between PsA and AS; the bacterial diversity decreases in PsA [32] and increases in AS [33].

## 4. IL-23

IL-23, consisting of a heterodimeric protein that contains subunits p19 and p40, was first cloned in 2000 [34]. IL-23 is produced by antigen-presenting cells (APCs) such as dendric cells (DCs), monocytes, and macrophages. Although IL-23 and IL-12 share subunit p40, only IL-12 can induce IFN-γ-producing Th1 cells. The increased production of IL-17 in response to IL-23 by CD4+ T cells was reported [35], and the novel population of IL-17-producing CD4+ T cells (Th17), clearly distinct from Th1 and Th2 cells, was identified in 2005 [36]. This discovery of the IL-23/Th17 pathway led to the clarification of the pathogenesis of autoimmune and autoinflammatory diseases that cannot be adequately explained by the Th1–Th2 concept [37]. Experimental evidence has demonstrated that the pathological activation of IL-23 and IL-17 triggers chronic inflammatory diseases, including SpA [38,39,40,41].

### IL-23 in SpA Patients

The upregulated presence of IL-23 in SpA patients confirms the involvement of the IL-23/IL-17 pathway. In AS, the elevated serum level of IL-17 and the expression of IL-23 p19 in peripheral blood mononuclear cells were demonstrated [42,43]. Additionally, increased expression of IL-23 in regional skin in patients with Ps and in synovial tissue in patients with PsA has been reported [44,45]. Despite the elevation of IL-23 levels in the synovial fluid in rheumatoid arthritis (RA) and PsA patients, only the disease activity parameters in RA patients correlated with IL-23 expression. These results suggest the existence of a different immunoregulation process for SpA and RA [46].

One question that arises here is where IL-23 originates. Two studies have reported the involvement of the intestine. Ciccia et al. found the upregulation of IL-23 expression, but not IL-17, in the terminal ileum in patients with AS and identified resident Paneth cells as a pivotal source of IL-23 production [47]. Furthermore, they found IL-23-responsive type 3 innate lymphoid cells (ILC3) in the gut and that IL-17+ ILC3 derived from the gut is expanded in the peripheral blood. These studies suggest that the intestine is at least one of the main producers of IL-23 [48].

## 5. IL-17

IL-17 is a pro-inflammatory cytokine that was first identified in 1993 [49]. The IL-17 cytokine family consists of six members, from IL-17A to IL-17F. IL-17A is the prototypical member, and IL-17A and IL-17F have the highest homology and overlap in many functions. There are five members of the IL-17 receptor family: IL17-RA, IL-17RB, IL-17RC, IL17-RD, and IL-17RE. IL-17 receptors exist as heterodimers, and IL-17RA is a common subunit. The IL-17 RA and IL-17RC heterodimer is the receptor of IL-17A, IL-17F, and the heterodimer of IL-17A and IL-17F. IL-17RA is ubiquitously expressed, whereas IL-17RC is mainly expressed on non-hematopoietic cells [50].

IL-17 plays an important role in acute inflammation by accumulating neutrophils through IL-6 [37]. IL-17R knockout mice displayed a significant delay in neutrophil recruitment and an attenuated host defense against bacterial infection [51,52], supporting the essential role of IL-17 in the inflammatory response. IL-17 targets various cells such as endothelial cells, fibroblasts, and macrophages, leading to the production of inflammatory cytokines [53,54] (Figure 2). The key role of IL-17 in the pathogenesis of chronic inflammatory diseases including SpA as well as in host defense has also been reported [54,55]. In addition to arthritis, IL-17 has been shown to affect bone metabolism by activating the production of matrix metalloproteinases by macrophages and the receptor activator of NF-κB ligand (RANKL) presented by osteoblasts [54,56]. Furthermore, IL-17 has direct effects on osteoclasts and activates osteoclastogenesis [56]. Thus, IL-17 has been demonstrated to be involved with the pathogenesis of SpA.

### 5.1. IL-17 in SpA Patients

Many studies have demonstrated the increased IL-17 production and mRNA expression in the serum, synovium, or tissue in RA patients [57]. These results indicate the importance of IL-17 in the development of arthritis. The increased IL-17 level was also reported in patients with SpA [43,58,59,60,61,62,63,64]. The serum levels of IL-17 were significantly higher in SpA patients (AS, re-SpA, and undifferentiated Spondyloarthritis [uSpA]) than in healthy controls [58,59] and were positively correlated to the disease activity measured by the Bath Ankylosing Spondylitis Disease Activity Index (BASDAI) [60,61]. In addition, the strong positive correlation between the serum levels of IL-17 and IL-23 in AS patients suggests the close relationship between these two cytokines [43].

As for the IL-17 levels in the patients’ tissues, a higher concentration of IL-17 in synovial fluid was reported in patients with PsA, re-SpA, and uSpA compared to patients with RA [62,63]. Furthermore, mRNA and protein levels of IL-17 receptor A were elevated in the synoviocytes of patients with PsA and RA compared to patients with osteoarthritis (OA) [64].

### 5.2. IL-17-Producing Cells

#### 5.2.1. Th17 Cells (IL-17+ CD4+ T Cells)

In 2005, a novel population of CD4+ T cells that secretes IL-17, clearly distinguished from Th1 and Th2, was identified [65,66]. IL-17 was initially recognized as a product of Th17 cells, but IL-17 production from other immune cells was also demonstrated [55].

IL-23 was thought to regulate the differentiation from CD4+ naive T cells to Th17 cells. But three independent studies revealed that IL-6 and TGF-β required to induce IL-17 in naïve T cells do not express IL-23 receptors [67,68,69]. Activation of STAT3 by IL-6 and TGF-β induces the master regulator of Th-17 cells, the transcription factor retinoid-related orphan receptor-γ (RORC), which expresses IL-23 receptor on the surface of Th 17 cells and enables them to secrete IL-17 in response to IL-23 [35]. IL-23 has now been shown to contribute to the lineage maintenance and proliferation of Th17.

The increased presence of Th17 in SpA patients has been reported. The serum levels of Th17 were higher in the peripheral blood in AS and PsA patients than in healthy controls [70,71]. Furthermore, several studies demonstrated the increased existence of Th17 in the inflamed tissue. The increase of Th17 cells in the synovial fluid in patients with PsA and reactive SpA has been reported [63,72,73]. As well as in SpA, the increased number of Th17 cells has also been identified in the skin of Ps patients, suggesting the involvement of local inflammation induced by Th17-derived IL-17 [74]. Immunohistological analysis of the facet joint in AS patients also revealed a higher expression of IL-17+ T cells [75]. 

In terms of the contribution of Th17 in the disease activity, correlations between the number of Th17 cells and disease activity score (BASDAI score) in AS [61] and the correlation between the number of Th17 cells in synovial fluid and C-reactive protein (CRP) level, erythrocyte sedimentation rate (ESR), and disease activity score (DAS) 28 in PsA were reported [72].

#### 5.2.2. γδT Cells

γδT cells, a subset of T cells that expresses γδTCR, account for a minor portion (3–5%) of all circulating T cells, but they are much more prevalent at mucosal and epithelial sites, where they account for approximately 50% of the intraepithelial lymphocyte population [76]. γδT cells play an important role in the mucosal defense against bacterial infection and antitumor immunity [77,78]. Among them, a subset of IL-17-producing γδT cells also play an especially pathological role in SpA [79]. In Ps patients, dermal γδT cells that produce IL-17 in response to IL-23 were increased in the affected skin, leading to disease progression [80,81]. In addition, the enrichment of circulating IL-17-secreting γδT cells was identified in patients with PsA and AS [82,83]. Cuthbert et al. identified tissue-resident γδT cells in the spinal enthesis, which were divided into two subsets: Vδ1 and Vδ2. While Vδ2 cells expressed IL-23/IL-17 axis-associated transcripts including IL-23R, Vδ1 completely lacked IL-23R expression and produced IL-17 in an IL-23-independent manner [84].

#### 5.2.3. Mucosa-Associated Invariant T (MAIT) Cells

Mucosal-associated invariant T (MAIT) cells are one of the subsets of innate-like cells, abundantly enriched in mucosal tissue, liver, and blood. MAIT cells act at the intersection of the innate and adaptive immune system and play an important role against bacterial infections. MAIT cells express the TCR αchain, and their activation depends on a restriction by a nonpolymorphic MHC-related molecule-1 (MR1) [85]. Through MR1 activation, MAIT cells produce pro-inflammatory cytokines such as IL-17, IL-22, and IFN-γ. Recent evidence has indicated the involvement of MAIT cells in the derivation of IL-17 in the Spondyloarthritis. IL-17-producing CD8+ MAIT cells were identified in psoriatic skin and blood in Ps patients [86] and were also identified in the synovial fluid in patients with PsA [72]. In addition, MAIT cells were found to be involved in AS. Although the frequency of circulating MAIT cells was lower than in healthy controls, the proportion of IL-17+ MAIT cells in the blood of patients with AS was elevated [87,88,89]. Higher proportions of MAIT cells were found in the synovial fluid than in the circulation. Hayashi et al. also demonstrated the correlation between the expression of CD69 on MAIT cells with the Ankylosing Spondylitis Disease Activity Score (ASDAS) in patients with AS. These results suggest that the upregulation of IL-17 by MAIT cells contributes to the pathogenesis of SpA.

#### 5.2.4. Type 3 Innate Lymphoid Cells (ILC3)

Innate lymphoid cells (ILCs) are immune cells that belong to the family of lymphoid effector cells and that play a pivotal role in immune defense, inflammation, and tissue remodeling [90]. ILCs are divided into three lineages based on their distinct production of cytokines: ILC1 produces IFN-γ; ILC2 is the predominant source of IL-4, IL-5, and IL-9; and ILC3 produces IL-17 and IL-22 in response to IL-23 [91].

Increased levels of IL-17-producing ILC3s, a lineage-negative cell population, have been identified in the peripheral blood of patients with PsA compared with healthy controls, and the levels of ILC3 correlated with the disease activity [92]. In AS patients, Cuthbert et al. identified ILC3s in the human enthesis [93] and Ciccia et al. demonstrated that gut-derived IL-17-producing ILC3s were expanded in the peripheral blood and synovial fluid [48]. 

#### 5.2.5. Other IL-17-Producing Cells

In 2005, a novel population of CD4+ T cells that secretes IL-17, clearly distinguished from Th1 and Th2, was identified [65,66]. IL-17 is initially recognized as a product of Th17 cells, but IL-17 production from other immune cells was also demonstrated [55]. 

The presence of IL-17-producing CD8+ T cells (also referred to as Tc17) has been identified in SpA patients. Increased levels of Tc17 were present in the peripheral blood of patients with AS, and the proportion of Tc17 positively correlated with the disease severity [94]. In addition, the increased number of Tc17 was identified in the synovial fluid in patients with PsA or AS [72,95].

Other IL-17-producing cells such as tissue-resident memory T cells, CD3-CD56+NK cells, and mast cells were also demonstrated in the skin, peripheral blood, or synovial fluid of patients with SpA [96,97,98]. However, further studies are needed to confirm the role of these cells in the pathogenesis of SpA.

## 6. IL-23/IL-17 Axis-Targeting Therapies

New targeted therapies using cytokine-specific monoclonal antibodies provide some of the most compelling evidence for the important roles of IL-23/IL-17 pathways in the pathogenesis of SpA. Various drugs along with IL-23 and IL-17 are summarized here (Table 1) and illustrated in Figure 3.

### 6.1. Anti IL-23

#### 6.1.1. Ustekinumab

Ustekinumab is a human monoclonal antibody that binds to the p40 subunit that is shared by both IL-12 and IL-23 and inhibits the functions of both IL-12 and IL-23. In three clinical trials in patients with PsA, ustekinumab treatment demonstrated higher improvement of the disease activity [99,100,101]. Following these results, ustekinumab received FDA approval for the treatment of PsA in 2013. On the other hand, there is no evidence of the efficacy of ustekinumab in the treatment of AS. Three placebo-controlled clinical trials of ustekinumab were conducted for AS. However, after the insufficient effectiveness of the first clinical trial, the following two trials were discontinued before full patient enrollment [102].

#### 6.1.2. Guselkumab

Guselkumab targets the p19 subunit that is shared between IL-23 and IL-39 [103]. While ustekinumab targets both IL-12 and IL-23 via binding to the p40 subunit, guselkumab inhibits IL-23 specifically. The results of two phase 3 clinical trials demonstrated the efficacy of guselkumab in the treatment of PsA [104,105]. The DISCOVER-1 trial aimed to assess the effects of guselkumab in patients with various levels of disease activity, while the DISCOVER-2 trial targeted biologic-naïve patients with active PsA. Guselkumab demonstrated greater improvement in the primary endpoint (American College of Rheumatology 20% improvement [ACR20] response at week 24) in both trials. These trials are being extended (DISCOVER-1 for 1 year and DISCOVER-2 for 2 years) to accumulate additional data related to the efficacy and safety of guselkumab [106].

#### 6.1.3. Tildrakizumab

Tildrakizumab is a high-affinity humanized antibody targeting IL-23 p19 and the second selective IL-23 antagonist and has been approved for the treatment of Ps by the FDA [107]. A phase 2b study for PsA reported the significant improvement of joint and skin manifestations. At week 24, patients receiving tildrakizumab achieved superior improvement in ACR20 and on the Psoriasis Area and Severity Index (PASI) 95 [108]. Furthermore, phase 3 trials are underway for PsA (NCT04314531 and NCT04314544) and AS (NCT-0355276). The most common treatment-emergent adverse events reported through week 24 were nasopharyngitis (tildrakizumab: 5.4%, placebo: 6.3%) and diarrhea (tildrakizumab: 1.3%, placebo: 0%).

#### 6.1.4. Risankizumab

Risankizumab is expected to have quick efficacy, based on the results of the phase 1 trial during which rapid and durable clearing of skin lesions was demonstrated in Ps patients [109]. Risankizumab is a humanized IgG1 monoclonal antibody that selectively inhibits the IL-23 p19 subunit. In the head-to-head trial of risankizumab and ustekinumab for Ps patients, risankizumab demonstrated significantly greater improvement in the PASI score at week 12, and this efficacy was maintained up to week 48 [110]. In this study, a 50% reduction of VAS score caused by arthritis was also reported in both the ustekinumab and risankizumab groups. 

As for PsA, a phase 2 trial demonstrated the primary end point of ACR20 response at week 16 was superior in the risankizumab-treated patients, and this efficacy was maintained at week 24 [111].

On the other hand, risankizumab demonstrated no significant effectiveness against active AS. A total of 159 patients with untreated active AS were included in the study. A roughly 40% improvement in Assessment in SpondyloArthritis International Society (ASAS40) score at week 12 was attained in 25.5%, 20.5%, and 15.0% of subjects in the 18 mg, 90 mg, and 180 mg risankizumab groups, respectively, compared with 17.5% of subjects in the placebo group [112].

### 6.2. Anti IL-17

#### 6.2.1. Secukinumab

Secukinumab is a fully human monoclonal IgG1 antibody that selectively inhibits IL-17A. After the phase 3 studies demonstrated its significant efficacy in 2015 [113,114], secukinumab received FDA approval for the treatment of AS and PsA in 2016. In other phase 3 trials, secukinumab also showed efficacy in delaying radiographic changes in AS patients whose illness was unable to be controlled by or was contraindicated by TNF-inhibitors [115], and a sustained, long-time efficacy in patients with AS was also reported [116,117]. With an extension of the phase 3 trial (FUTURE 2) [114], the long-term effects of secukinumab in PsA were also demonstrated [118,119]. In addition, a head-to-head comparison study of secukinumab and adalimumab involving a first-line biological monotherapy (EXCEED) trial demonstrated that the ACR 20 response at week 52 was 67% in the secukinumab group and 62% in the adalimumab group (OR, 1.30; 95%CI, 0.98–1.72; *p* = 0.0719). Although the difference was not statistically significant in terms of the ACR20 response, a higher treatment retention rate and better improvement in PASI was reported, which suggests that secukinumab is a useful option for the treatment of AS [120]. 

#### 6.2.2. Ixekizumab

Ixekizumab is a humanized IgG4 monoclonal antibody that neutralizes IL-17A in contrast to secukinumab, which is an IgG1 monoclonal antibody. This structural difference characterizes the higher affinity of ixekizumab to IL-17 [121]. For the treatment of Ps, ixekizumab showed a superior short-term outcome compared to secukinumab [122,123]. In a phase 3 clinical trial (SPRIT-P1) [124], ACR20 response at week 24 was superior in the ixekizumab group (60.3%) compared to placebo (30.1%), and an extension of the study revealed the long-term persistent efficacy, safety, and inhibition of radiographic progression [125]. As for the treatment of AS, ixekizumab also demonstrated a superior response of ASAS40 at week 16 compared to placebo (ixekizumab: 52%, placebo: 18%) [126]. Ixekizumab received approval by the FDA for the treatment of PsA in 2017, radiographic axial SpA in 2019, and non-radiographic axial SpA in 2020. The comparison between ixekizumab and adalimumab in the treatment of PsA demonstrated that the effects related to joint improvement in ACR20/50/70 response were comparable, but ixekizumab had a greater response for skin manifestations (PASI) [127].

#### 6.2.3. Netakimab

Netakimab (BCD-085) is a novel recombinant IgG1 anti-IL-17 monoclonal antibody with a modified complementarity determining region (CDR) and Fc-fragment, which results in a higher affinity of IL-17 to the Fab fragment of netakimab [128,129].

A phase 3 trial for PsA (NCT03598751: PATERA study) [130,131] is in progress and is expected to generate new evidence in support of the use of netakimab in SpA treatment.

#### 6.2.4. Brodalumab

IL-17 receptor (IL-17R) is a heterodimer of IL17-RA and IL-17RC. Brodalumab is a humanized IgG2 monoclonal antibody that binds to IL-17RA and inhibits IL-17A, IL-17A/F, IL-17F/F, and IL-17E [132]. A phase 2 clinical study on PsA patients demonstrated that ACR20 response rates at week 12 in the 140 mg and 280 mg brodalumab groups were 37% and 39%, respectively, as compared with 18% among the placebo group [133]. A study with brodalumab for AS was discontinued because of suicidality being one of the adverse events that developed during the trial [134] although no causal relationship between suicidality and brodalumab treatment was confirmed [135].

#### 6.2.5. Bimekizumab

Bimekizumab is a humanized IgG1 monoclonal antibody with dual neutralization effects of both IL-17A and IL-17F [136]. A phase 2b dose-ranging trial (BE ACTIVE study) [137] and one for AS (BE AGILE study) [138] demonstrated the short-term efficacy of bimekizumab for PsA. In the BE ACTIVE study, the ACR50 response at week 12 was greater in the 16 mg (27%), 160 mg (41%), and 320 mg bimekizumab (24%) groups than in the placebo group (7%) [137]. In addition, the primary end point (ASAS40 at week 12) was achieved in all of the bimekizumab-treated groups (16 mg: 29.5%, 64 mg: 42.6%, 160 mg: 46.7%, and 320 mg: 45.9%) compared with placebo (13.3%) [138].

### 6.3. Bispecific TNF/IL-17A Inhibitor

#### ABT-122

ABT-122 is an IgG1 dual-variable domain immunoglobulin (DVD-Ig) that was engineered to bind to and neutralize human TNF and IL-17A. It is built on an adalimumab backbone by adding IL-17A binding domains [139,140,141]. Although the dual inhibition of TNF and IL-17A was expected to provide greater effects compared to adalimumab, which targets only TNF, two phase 2 clinical studies have failed to demonstrate the superiority of ABT-122 for the treatment of PsA compared to adalimumab at 12 weeks [141,142]. 

## 7. Future Prospective

While the targeting IL-23 and IL-17 have shown good results for the patients with AS and PsA, it was also effective in the treatment of IBD [143,144]. This finding suggests that the IL-23/ IL-17 pathway isn’t specific in the development of SpA

There remains an unclear mechanism that is not fully explained by only IL-23 and IL-17 in the pathogenesis of SpA. According to the results of clinical trials, anti-IL-23 antibody showed significant efficacy in the patients with peripheral SpA, but not in the patients with AS. Besides, only IL-23 and IL-17 cannot explain the different sites of manifestations between axial SpA and peripheral SpA, implying the contribution of the mechanisms that are independent of the IL-23/IL-17 pathway. However, different action mechanisms of IL-23/IL-17 pathway between axial SpA and peripheral SpA remains unknown.

The evidence has been insufficient about the efficacy of targeting IL-23 and IL-17 for the patients with other subtypes of SpA, such as re-SpA, IBD-associated spondylitis, and unclassifiable spondylitis. 

Further research is required to elucidate underlying mechanism for the difference between axial SpA and peripheral SpA including the mechanism other than IL-23/IL-17 pathway and to evaluate the therapeutic effects of targeting the IL-23/IL-17 in each subtype of SpA.

## 8. Conclusions

Recent studies have clarified the essential roles of IL-23 and IL-17 in SpA. Although clinical and genetic evidence has demonstrated the involvement of IL-23 and IL-17 in the various tissues from patients with SpA, this IL-23/IL-17 pathway cannot explain the whole range of pathogenic conditions in SpA, which suggests the involvement of unknown pathways or mechanisms for the development of SpA. IL-17 originally acts only during an immune response, yet genetic and environmental conditions such as HLA-B27, mechanical stress, and dysbiosis can cause a pathological up-regulation of IL-23 and/or IL-17 and lead to the development of SpA. Indeed, specific inhibition of IL-23 and/or IL-17 by monoclonal antibodies have shown the effects on the control of the disease activity of SpA. The IL-17-producing cells such as Th17, Tc17, and γδT cells and ILC3 were newly identified. However, whether the IL-17 from different cellular sources exerts a different immunopathological response remains to be determined. Further understanding of cellular and molecular regulatory mechanisms of the IL-23/IL-17 axis and other inflammatory cytokines may provide a promising strategy in the treatment of SpA.

## Figures and Tables

**Figure 1 ijms-21-06401-f001:**
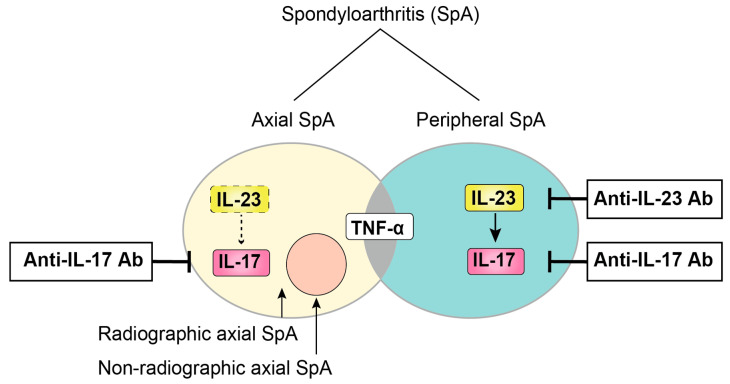
Schematic classification of SpA. SpA patients are mainly classified into two groups (axial SpA and peripheral SpA) based on their predominant clinical manifestation, with some overlap between these two groups. Axial SpA are further divided into two subtypes depending on whether there is radiographical bone destruction.

**Figure 2 ijms-21-06401-f002:**
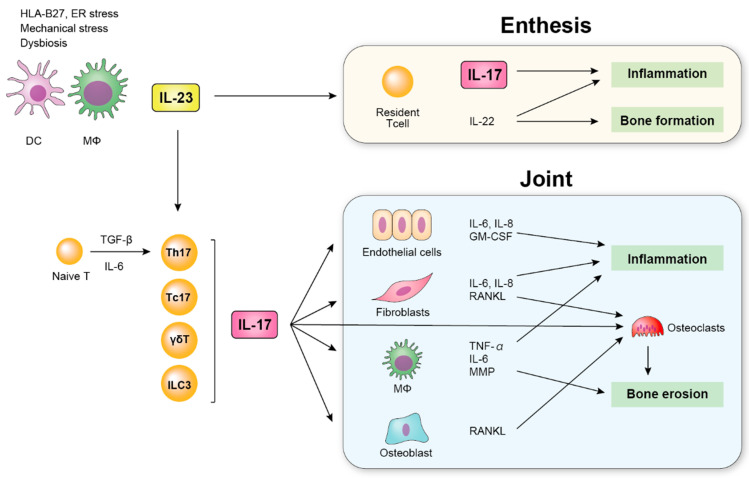
Schematic overview of the role of IL-23 and IL-17 in SpA. Dendric cells (DCs) and macrophages (MΦs) produce IL-23. IL-23 induces the production of IL-17 by various cells and contributes to inflammation by upregulating the production of inflammatory cytokines such as IL-6 and TNF-α, which induces inflammation in the enthesis and joint.

**Figure 3 ijms-21-06401-f003:**
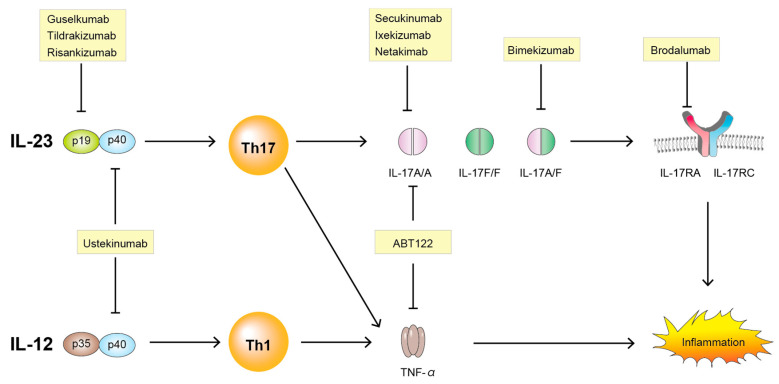
Targeting the IL-23/IL-17 pathway. IL-23, interleukin-23; IL-12, interleukin-12; p19, p19 subunit of interleukin-23; p35, p35 subunit of interleukin-12; p40, p40 subunit of interleukin-12 or of interleukin-23; IL-17A, interleukin-17A; IL-17F, interleukin17F; IL-17RA, interleukin-17 receptor A; IL-17RC, interleukin-17 receptor C; Th1, T helper cell 1; Th17, T helper cell 17; TNF-α, tumor necrosis factor-α.

**Table 1 ijms-21-06401-t001:** Agents targeting IL-23/IL-17 pathway.

Name	Target Cytokine	AS	PsA	Other Autoinflammatory Diseases
Ustekinumab	IL-12-p40 and IL-23-p40	Phase 3	Marketed	CD,UC (Marketed)
Guselkumab	IL-23-p19	-	Marketed	CD,UC (Phase 3)
Tildrakizumab	IL-23-p19	Phase 2	Marketed	-
Risankizumab	IL-23-p19	Phase 2	Marketed	CD,UC (Phase 3)
Secukinumab	IL-17A	Marketed	Marketed	-
Ixekizumab	IL-17A	Marketed	Marketed	-
Netakimab	IL-17A	Phase 3	Phase 3	-
Brodalumab	IL-17RA	Phase 3	Phase 3	CD (Phase 2)
Bimekizumab	IL-17A/F	Phase 2	Phase 3	-
ABT-122	IL-17A and TNF-α	-	Phase 2	-

CD; Crohn’s disease, UC; Ulcerative colitis.

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
