# Peer review of "The Role of the IL-23/IL-17 Pathway in the Pathogenesis of Spondyloarthritis"

_ijms, 2020, doi:10.3390/ijms21176401_

Round 1

Reviewer 1 Report

Authors reviewed the roles of IL 23/17 on chronic inflammmation such as SpA and targeting therapies.

It's understandable and profitable for the related researchers and clinicians.  

Author Response

Thank you very much for your complaisant comment.

Reviewer 2 Report

To authors (IJMS_905459),

This paper reviewed previous studies focusing on spondyloarthritis (SpA) pathology by IL-23/IL-17 pathway and drugs modifying the pathogenesis. Overall, the manuscript is well written, concise, and covered general SpA pathology, which has been already cited in previous similar review papers.  

However, there are some minor issues to be addressed to make the purpose of the review clear.

  1. It has been well known that the IL-23/IL-17 pathway is important for the pathogenesis of immune-mediated several inflammatory diseases. In this paper, the author focused on the pathogenesis of SpA by IL-23/IL-17 with describing different two subtypes such as axial SpA and peripheral SpA, however it doesn’t seem that the author discusses more distinguishable action mechanisms of IL-23 and / or IL-17 between axial SpA and peripheral SpA. It would be better if the author review more specific molecular mechanisms of IL-23 and / or IL-17 that result in the axial SpA and peripheral SpA, respectively.
  2. It seems better understand if target cytokines such as IL-23 and/or IL-17 are briefly illustrated in the figure showing a classification of SpA.      
  3. In line 341, the author makes sure that the drug, Guselkumab, is completely duplicated with the Anti-IL-23 in line 288
  4. The author described various approved inhibitors and illustrated target cytokines of these inhibitors in Figure 3. Which is great, however it would be good too if the inhibitors are simply summarized based on target cytokine and target SpA as a table.

Reviewer 3 Report

In this review "The role of the IL-23/IL-17 pathway in the pathogenesis of spondyloarthritis". The authors summerazed the role of IL-23/IL-17 pathway in the SpA. This review provide the information of pathogenesis of SpA.

Minor points:

  1. The further development direction should be discussed.
  2. The authors should discuss the limintation of IL-23/IL-17 pathway.
  3. The possible application of IL-23/IL-17 in other diseases should be discussed.

Reviewer 4 Report

This review is well written.The content on SpA was well organized and widely described.
